# Brains vs. Bytes: Evaluating LLM Proficiency in Olympiad Mathematics

**Hamed Mahdavi, Pegah Mohammadipour, Samira Malek, ,Vasant Honavar**
Pennsylvania State University
University Park, PA, USA
{hmm5834,pegahmp,sxm6547,vhonavar}@psu.edu

**Alireza Hashemi**
City University of New York
New York, NY, USA
alireza.hashemi13@outlook.com

**Majid Daliri**
New York University
New York, NY, USA
daliri.majid@nyu.edu

**Alireza Farhadi**
Amirkabir University of Technology
Tehran, Iran
farhadi@aut.ac.ir

**Yekta Yazdanifard**
Bocconi University
Milan, Italy
yekta.yazdanifard@unibocconi.it

**Amir Khasahmadi**
Autodesk
Toronto, Canada
amir.khasahmadi@autodesk.com

## Abstract

Recent advancements in large language models (LLMs) have shown impressive progress in mathematical reasoning tasks. However, current evaluation benchmarks predominantly focus on the accuracy of final answers, often overlooking the logical rigor crucial for mathematical problem-solving. The claim that state-of-the-art LLMs can solve Math Olympiad-level problems requires closer examination. To explore this, we conducted both qualitative and quantitative human evaluations of proofs generated by LLMs, and developed a schema for automatically assessing their reasoning capabilities. Our study reveals that current LLMs fall significantly short of solving challenging Olympiad-level problems and frequently fail to distinguish correct mathematical reasoning from clearly flawed solutions. We also found that occasional correct final answers provided by LLMs often result from pattern recognition or heuristic shortcuts rather than genuine mathematical reasoning. These findings underscore the substantial gap between LLM performance and human expertise in advanced mathematical reasoning and highlight the importance of developing benchmarks that prioritize the rigor and coherence of mathematical arguments rather than merely the correctness of final answers.

## 1 Introduction

The release of OpenAI's o1 model (OpenAI et al., 2024) marks a significant breakthrough in artificial intelligence research, particularly in the domains of reasoning and problem solving. Building on this achievement, several state-of-the-art models have been introduced (DeepSeek-AI et al., 2024; Qwen et al., 2025; Google, 2025), which incorporate post-training on chain-of-thought data. These models demonstrate enhanced capabilities in reasoning and solving mathematical problems. Although the exact methodologies and best practices for developing reasoning models remain an active area of research, the use of post-training techniques has proven pivotal in improving performance on tasks requiring planning, iterative thinking, and trial-and-error strategies. By generating reasoning tokens before producing a final answer, these models offer more deliberate and reliable solutions, particularly for complex reasoning tasks such as mathematical problem-solving.

A range of different benchmarks, such as GSM8K (Cobbe et al., 2021) and MATH (Hendrycks et al., 2021), have been developed to evaluate the capability of large language models (LLMs) to address mathematical challenges. As these models advance in performance—e.g., Qwen2.5-72B achieving 91.5% accuracy on GSM8K and 62.1% on MATH, while OpenAI's O1 attains 94.8% on the MATH dataset—more challenging benchmarks have been introduced. These include OlympiadBench (He et al., 2024), OlympicArena (Huang et al., 2024b), CHAMP (Mao et al., 2024), AlphaGeometry (Trinh et al., 2024), MathOdyssey (Fang et al., 2024), and Omni-MATH (Gao et al., 2024), which contain contest-level mathematical problems designed to push the boundaries of LLM capabilities.

Except for CHAMP, which includes "concepts, general math facts, and hints" (Mao et al., 2024) as additional annotations, and AlphaGeometry, which is distinct as it contains problems and solutions in their formal translation form, all other benchmarks rely on final answer correctness, either symbolic or numerical, as the evaluation metric to assess the reasoning capabilities of various language models.

This approach presents a clear issue: Models may exploit heuristics or flawed reasoning yet still arrive at the correct final answer, rather than relying on logically sound methods. This limitation is less evident in widely-used reasoning benchmarks like GSM8K and MATH, which mainly include simpler problems where finding the correct solution is often equivalent to solving the problem correctly. In contrast, benchmarks featuring contest-level problems—such as CHAMP, OlympiadBench, Omni-MATH, MathOdyssey, and OlympicArena—require greater emphasis on the validity and soundness of the solution's reasoning, not just the accuracy of the final answer. This distinction becomes particularly crucial when addressing open-ended questions.

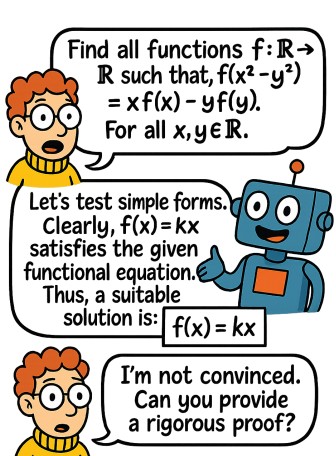

Figure 1: LLMs often fail to generate logically sound solutions for challenging problems.

LLM-as-a-Judge is a recent method to automatically assess the quality of LLM-generated responses (Li et al., 2025; 2024; Cohen et al., 2023; Adlakha et al., 2024; Liu et al., 2023; Manakul et al., 2023). Directly evaluating LLM-generated solutions, as in Dubois et al. (2023), can be inherently complex for challenging math problems, as verifying the correctness of a solution may itself require significant effort. Applying consistent rubrics for automatic grading of generated solutions is also nontrivial because the relative importance of different facts may vary across solutions and problems. In the context of math Olympiad problems, some mistakes are fundamental and reflect deeper misunderstandings, while others are more superficial and could be fixed with minor adjustments. There is no clear, objective method to define which mistakes are critical and which are less significant. Sawada et al. (2023) proposes a rubric-based approach where a reference solution and evaluation criteria are used to assess the quality of a generated solution. However, this approach faces an obvious limitation: the existence of multiple valid solutions that may not align with the ref-

erence solutions. Furthermore, the challenges inherent in directly assessing LLM-generated solutions also persist with this approach.

Given these limitations, our goal in this work is twofold: first, to evaluate the quality of LLM-generated solutions to mathematics Olympiad problems, including identifying their common failure modes and error types; and second, to investigate whether LLMs can verify the correctness of their own solutions. This study is significant for two main reasons. First, evaluating the correctness of the proofs generated by LLMs can reveal whether training strategies that rely solely on the correctness of the final answer are sufficient for guiding LLMs toward producing valid and logically sound proofs. Second, assessing the ability of LLMs to verify their own generated solutions can provide insights into the feasibility of bootstrapping verification processes, potentially enabling LLMs to improve the quality of their proofs.

To achieve our goal, we assembled a group of evaluators to assess the quality of LLM-generated solutions for problems from the International Mathematics Olympiad (IMO) shortlist. After carefully analyzing solutions generated by the frontier models such as o1, o1-mini, o3-mini, Gemini 2.0 (Flash Thinking mode), and DeepSeek R1[1], we observed that only a negligible percentage of these solutions were correct or provided meaningful, non-trivial insights into the problems. We found that LLM-generated solutions frequently contained common types of fallacies, which made their reasoning blatantly incorrect. To address this, we defined and categorized these fallacies, then systematically evaluated solutions generated by frontier models, analyzing their correctness and labeling each incorrect solution by fallacy type. We assessed the ability of frontier models to verify solutions. Our findings revealed that, in nearly all cases, the models failed to distinguish between correct solutions and incorrect ones containing obvious fallacious arguments. Importantly, identifying such naive fallacies is a simpler task than full solution verification, yet the models still struggled to perform reliably. In summary, our contributions are as follows:

- We conducted an extensive evaluation of solutions generated by frontier models on 455 IMO shortlist problems, emphasizing proof correctness rather than merely checking final answers.
- To gain deeper insight into recurring mistakes made by LLMs, we systematically identified and categorized common logical fallacies present in their solutions, establishing a comprehensive framework for classifying typical errors.
- Leveraging these findings, we created a labeled dataset by annotating each solution according to correctness and the type of fallacy exhibited. This dataset supports tasks related to solution verification and offers valuable insights into the current capabilities and limitations of the frontier LLMs.
- Our analysis reveals that even advanced models frequently struggle to distinguish between valid solutions and those containing evident logical fallacies.

## 2  Related Work

**Benchmarks:** Various datasets evaluate mathematical reasoning in large language models (LLMs) (Ahn et al., 2024). Some focus purely on arithmetic problems (Yuan et al., 2023), while math word problem (MWP) datasets, like GSM8K (Cobbe et al., 2021) and MathQA (Amini et al., 2019), present natural language scenarios requiring logical reasoning (Wei et al., 2023).

Recently, the limitations of LLMs on seemingly simple problems have motivated benchmarks designed to probe robustness and compositionality. GSM1K (Zhang et al., 2024) carefully examines model performance on adversarial arithmetic, revealing surprising fragility even on variations closely related to GSM8K. Building on this, Compositional GSM (Hosseini et al., 2024) introduces multi-step arithmetic tasks requiring compositional

---

[1]When we started this project, models like Claude 3.7, o3-mini, Gemini 2.5, and Grok 3 hadn't been released yet, so we could only include o3-mini in our evaluations. After the submission process, we conducted evaluations of Gemini 2.5 Pro and have added them to the appendix.

reasoning, further highlighting challenges in generalization. Functional MATH (Srivastava et al., 2024) expands the landscape by evaluating systematic generalization and reasoning robustness via functionally diverse tasks, pushing LLMs beyond pattern matching and toward flexible abstraction.

Automated theorem proving (ATP) datasets evaluate models' capabilities in logical theorem proving (Zheng et al., 2022; Yu et al., 2024; Jiang et al., 2024). Recent benchmarks focus on advanced or Olympiad-level mathematics, such as CONIC10K for conic sections (Wu et al., 2023), GHOSTS and miniGHOSTS for graduate-level mathematics (Frieder et al., 2023), and CHAMP (Mao et al., 2024), OlympiadBench (He et al., 2024), MathOdyssey (Fang et al., 2024), and Omni-MATH (Gao et al., 2024), specifically focusing on competition-level problems. HARP (Yue et al., 2024) provides human-annotated US competition problems, and NuminaMath offers a large-scale collection of math problems and solutions (LI et al., 2024).

**LLM-as-a-judge:** Utilizing large language models as evaluative judges has gained popularity, reducing reliance on human annotations (Stephan et al., 2024; Li et al., 2024; Nasrabadi, 2024; Ning et al., 2024). This paradigm offers adaptable evaluations based on task-specific contexts (Tan et al., 2024; Dhurandhar et al., 2024), and its effectiveness is typically measured against human judgments (Kim et al., 2024; Ye et al., 2024; Liu et al., 2025). Benchmarks like UltraFeedback (Cui et al., 2024), AlpacaEval (Dubois et al., 2024), Chatbot Arena (Chiang et al., 2024), and MT-Bench (Zheng et al., 2023) evaluate different LLM judging domains. Specifically for mathematical reasoning, REASONEVAL (Xia et al., 2025) assesses answer correctness and reasoning validity, while MATHCHECK (Zhou et al., 2024) uses LLMs for robust evaluation across diverse mathematical tasks. The SMART-840 dataset (Cherian et al., 2024) benchmarks zero-shot mathematical reasoning based on human performance statistics.

**Mathematical Reasoning in LLMs:** Large language models (LLMs) have shown success in various reasoning tasks, especially when employing prompting techniques like Chain-of-Thought (CoT), which encourages them to generate correct intermediate steps toward a solution (Chen et al., 2024; Wei et al., 2023; Kojima et al., 2023). These methods can significantly boost performance on challenging problems (Havrilla et al., 2024). Furthermore, inference-time techniques like CoT with Self-Consistency (CoT-SC) have been developed to enhance reasoning by generating multiple reasoning paths and selecting the most consistent one (Wang et al., 2023; Wang & Zhou, 2024). Benchmarks like MATH (Hendrycks et al., 2021), GSM-Symbolic and GSM-NoOp (Mirzadeh et al., 2024) have been introduced to provide more controllable evaluations and reveal limitations such as sensitivity to numerical variations and irrelevant information, suggesting a potential lack of deep understanding of mathematical concepts. These benchmarks show that current LLMs rely more on probabilistic pattern-matching than genuine formal logical reasoning. To further refine LLMs' reasoning, approaches like reward modeling to evaluate solution correctness and self-refinement techniques (Huang et al., 2024a) and decomposing problems into smaller, algorithmic steps (Zelikman et al., 2023) are being explored.

## 3 General Workflow

In this section, we outline the data collection process. In the first phase, we selected a set of challenging problems to evaluate the quality of solutions generated by the LLMs. We gathered a group of seven evaluators, each either a former national-level Olympiad medalist or holding or doing a relevant PhD in fields like mathematics or computer science, and asked them to analyze the correctness of the LLM-generated solutions.

### 3.1 Problem Selection Rationale

Through this project, we used IMO shortlist problems. The process of selecting problems for the IMO Shortlist is rigorous and carefully coordinated. Each participating country submits a set of candidate problems, which typically span the main four mathematical fields: algebra, geometry, combinatorics, and number theory. These submissions are reviewed

by a problem selection committee to ensure they meet key criteria, including originality, mathematical depth, and suitability for the competition. The committee carefully evaluates the problems for their difficulty level, ensuring a balance between accessibility for less experienced participants and sufficient challenge for the most advanced contestants. From this review process, a shortlist is created, containing a diverse collection of high-quality problems. This shortlist forms the basis for the final selection of problems used in the IMO.

The shortlist problems have some distinct features that make them suitable for testing the mathematical reasoning capabilities of the frontier models:

- Shortlist problems are highly original, even within the context of contest-level problems. The selection committee ensures that the solution ideas for these problems are as novel and unique as possible. As a result, while attempting to generate solutions, an LLM cannot simply combine standard building blocks from well-known problems to arrive at the correct answer.

- It is almost always verified whether the problem can be reduced to a well-established result in advanced mathematics, such as undergraduate or graduate-level topics or research-level findings. Consequently, an LLM cannot leverage its extensive knowledge to apply an advanced mathematical result for an easy solution.

- All problems are designed to be solvable using high-school level mathematics, with the challenge lying in the intricacy of the ideas rather than requiring a background in advanced mathematics.

- The solutions typically involve multiple steps, each requiring nuanced arguments. Solving these problems demands careful planning, systematic thinking, and rigorous verification of each step, in contrast to simpler mathematical problems that can be tackled through straightforward algebraic manipulations or trial and error.

On the other hand, since IMO and IMO Shortlist problems are highly reputable, there is a significant likelihood of data leakage, as frontier models may have been trained on publicly available high-quality mathematical datasets. In this paper, we demonstrate that even if such data leakage has occurred, it does not substantially affect the ability of the LLM to solve IMO Shortlist-level problems.

## 3.2 Classifying the Failure Modes of LLM Solutions

We asked evaluators to present several IMO shortlist and shortlist-level problems to frontier models, including OpenAI o1, o1-mini, o3-mini, DeepSeek R1, and Gemini 2, and qualitatively analyze the details of the LLM-generated arguments. We found out that when these frontier models generate incorrect solutions, the errors consistently follow common patterns. Specifically, incorrect solutions typically involve blatantly inaccurate mathematical arguments or statements.

After thorough analysis, we identified the following fallacies commonly occurring in incorrect LLM-generated solutions. To illustrate each error type, demonstrative as well as real examples are provided in the appendix.

**Proof by Example.** Drawing a general conclusion based on a limited number of specific instances without rigorous justification for all cases. This error occurs when a statement appears valid in a few examples, misleadingly suggesting universal validity.

**Proposal Without Verification.** Introducing a method or strategy without properly justifying its correctness. The model proposes an idea but provides no rigorous argument or proof supporting its validity.

**Inventing Wrong Facts.** Citing or inventing non-existent theorems, definitions, or facts to justify a claim. Instead of relying on established mathematical facts, the argument relies on fabricated statements.

**Begging the Question (Circular Reasoning).** Assuming the conclusion's truth within the argument itself, thereby creating inherently flawed logical reasoning.

**Solution by Trial-and-Error.** Offering solutions derived solely from guesswork or testing a few random examples without providing reasoning for why selected solutions work or why alternatives are not considered.

**Calculation Mistakes.** Committing substantial arithmetic or algebraic mistakes that critically undermine the overall correctness of the solution. We specifically considered calculation errors severe enough to compromise the validity of the conclusion.

### 3.3 Data Annotation Processs

After defining the fallacy categories, we provided a list of IMO shortlist problems and corresponding model-generated solutions to the evaluators, instructing them to classify these solutions based on their correctness. We employed the following checklist to annotate our data:

1. First, the evaluator read the solution and determined whether it was correct, partially correct, or incorrect.
   - A solution was considered *correct* if it fully addressed all aspects of the problem and contained no significant errors in statements or conclusions.
   - A solution was deemed *partially correct* if it included some essential steps of a correct solution but omitted other crucial steps or contained significant inaccuracies.
   - A solution was classified as *incorrect* if it lacked any non-trivial useful information relevant to solving the problem.
2. If the solution was not correct, the evaluator identified the categories of fallacies present. In some cases, multiple fallacies could be identified.
3. If the problem required a final answer, the evaluator recorded both the correct final answer and the model-generated final answer.

To ensure consistency, the evaluators' team lead conducted a thorough review of the evaluators' outputs in parallel, verifying that the definitions of fallacies were applied correctly and consistently across all evaluators. Borderline cases were identified and discussed separately.

## 4 Human Evaluation Results

We evaluated the models using IMO shortlist problem sets from the years 2009 to 2023, comprising a total of 455 problems. These included 108 algebra, 117 combinatorics, 116 geometry, and 114 number theory problems. The number of problems per annual shortlist varied slightly, typically ranging from 26 to 35. Each set was carefully curated to maintain a balanced distribution of difficulty across the four primary mathematical topics. The performance of each model on the IMO shortlist problems is summarized in Table 1. As evident from the results, none of these models achieve performance levels comparable to those obtained through calculating final answers' accuracy, as reported in Gao et al. (2024) and Fang et al. (2024).

| Model | Correct (%) | Partially Correct (%) | Incorrect (%) |
|-------|-------------|-----------------------|---------------|
| DeepSeek | 3.8 | 6.7 | 89.4 |
| Gemini 2.0 | 0.0 | 1.1 | 98.9 |
| o1 | 1.9 | 3.9 | 94.2 |
| o1-mini | 0.0 | 0.0 | 100.0 |
| o3-mini | 3.3 | 4.4 | 92.2 |

Table 1: Performance of different models on IMO shortlist problems (%)

The observed gap between the outcomes of our evaluation and other methods that focus exclusively on the correctness of final answers arises because models may produce incorrect

intermediate steps yet still obtain the correct final result. To investigate this issue, we specifically examined problems with concrete final answers within our evaluation set. Table 2 illustrates both the proportion of correct final answers and the conditional probability of having a correct solution, given that the final answer is correct. Interestingly, we found that the frontier models still predominantly generated wrong solutions despite arriving at the correct final answer.

| Model | Final Answer Accuracy (%) | Correct\|Correct Final Answer (%) |
|---|---|---|
| DeepSeek | 63.2% | 0% |
| Gemini 2.0 | 43.8% | 0% |
| o1 | 30.8% | 12.5% |
| o1-mini | 35.0% | 0% |
| o3-mini | 48.3% | 14.3% |

Table 2: Comparison of evaluated LLMs highlighting the gap between final answer correctness and overall solution quality. **Final Answer Accuracy** denotes the percentage of correct final answers, whereas **Correct|Correct Final Answer** represents the percentage of fully correct solutions among instances where the final answer is correct.

These results suggest that the models rely on heuristics, shortcuts, and educated guesses rather than constructing logically sound solutions. Such strategies are particularly applicable for Olympiad-level problems, where the final answers often exhibit predictable patterns or can be inferred without deriving a complete solution. As a result, we argue that evaluating model performance based solely on final answer accuracy is fundamentally flawed, as it ignores the critical problem of solution validity and the systematic exploitation of answer predictability.

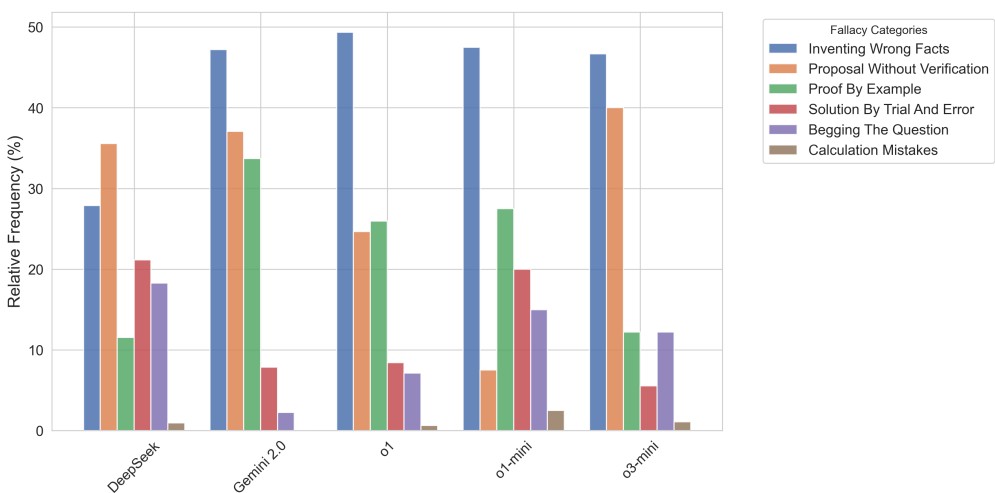

Figure 2: Relative frequencies of each fallacy among the LLM-generated solutions for each model.

It is also helpful to examine how common fallacies are distributed among different models. Figure 2 shows the relative frequency of each fallacy found in the LLM-generated solutions from each model. Relative frequencies are used because multiple fallacies may occur within a single solution. Among these fallacies, *Inventing Wrong Facts* is the most frequent in four models and the second-most frequent in another. This observation might be explained by the training methods used for these models, which generally involve reinforcement learning algorithms with rewards based on the correctness of the final answers (e.g., see the DeepSeek report; (DeepSeek-AI et al., 2024)).

*Proposal Without Verification* is the other common fallacy. Although the internal reasoning tokens are not visible for OpenAI models, an examination of the thinking traces from DeepSeek and Gemini suggests that this fallacy often arises because the model struggles to determine which calculations and statements should be included in the final response. As a result, instead of presenting concrete reasoning or useful intermediate steps, the model may produce vague mathematical claims, often beginning with phrases like "It is easy to show that..." or "One can show that..." without providing a proper supporting argument.

We also observed that models tend to exhibit different types of fallacious reasoning depending on the problem type. Figure 3 illustrates the relative frequencies of various fallacies in questions with and without final answers. Notably, *Proof by Example* and *Solution by Trial and Error* occur more frequently in questions where a final answer is provided. This suggests that models often arrive at final answers either through heuristic trial and error or by generalizing from a small number of test cases, leading to a higher prevalence of these two types of fallacies in their generated solutions. In contrast, we observe that *Inventing Wrong Facts* and *Proposal Without Verification* occur more frequently in generated solutions to problems lacking explicit final answers. Since these problems are purely proof-based, it is natural to observe these particular fallacies more often. To produce valid proofs, models must logically connect the problem's initial assumptions and constraints to the required conclusion. However, we find that models frequently circumvent this rigorous reasoning process either by introducing incorrect statements or by omitting essential steps in their arguments.

Similar patterns of difference can be observed in relative frequencies of fallacies among different problem topics. Figure 4 demonstrates the relative frequencies of fallacies among geometry, algebra, combinatorics, and number theory problems. As more geometry questions can be solved only using logical statements rather than algebraic manipulations, *Inventing Wrong Facts*, *Proposal Without Verification*, and *Begging the Question* are more common in geometry problems. A significant proportion of algebra problems fall into categories such as functional equations, polynomial equations, or optimization tasks. We observed that

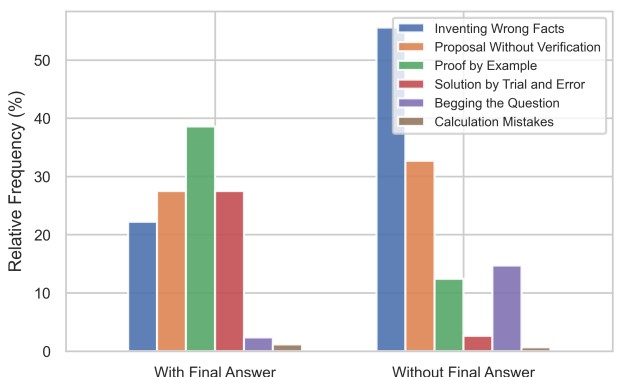

Figure 3: Relative frequencies of each fallacy in LLM-generated solutions, comparing questions with and without a final answer.

all frontier models tend to avoid generating rigorous analytic solutions, instead relying on trial and error to determine the final answer. This behavior results in a higher frequency of the *Solution by Trial and Error* fallacy in LLM-generated solutions for algebra problems. Similarly, number theory problems involving Diophantine equations or integer-valued functional equations exhibit the same issue. Additionally, we found that the *Proof by Example* fallacy occurs more frequently in algebra, combinatorics, and number theory problems compared to geometry. This trend arises because many problems in these three areas can be framed as proving statements of the form $Q(x)$, where $x$ belongs to a specific domain defined by the problem. In such cases, LLMs frequently attempt to verify the proposition $Q$ by evaluating selected examples from its domain rather than constructing a general proof, thus resulting in the *Proof by Example* fallacy.

## 5 Automatic Evaluation Results with Gemini 2.5 Pro

Verification of a candidate solution to a problem is generally considered an easier task than solving the problem itself. Consequently, a common strategy for training reasoning-oriented LLMs is the generator-verifier schema. Within this framework, a generator produces

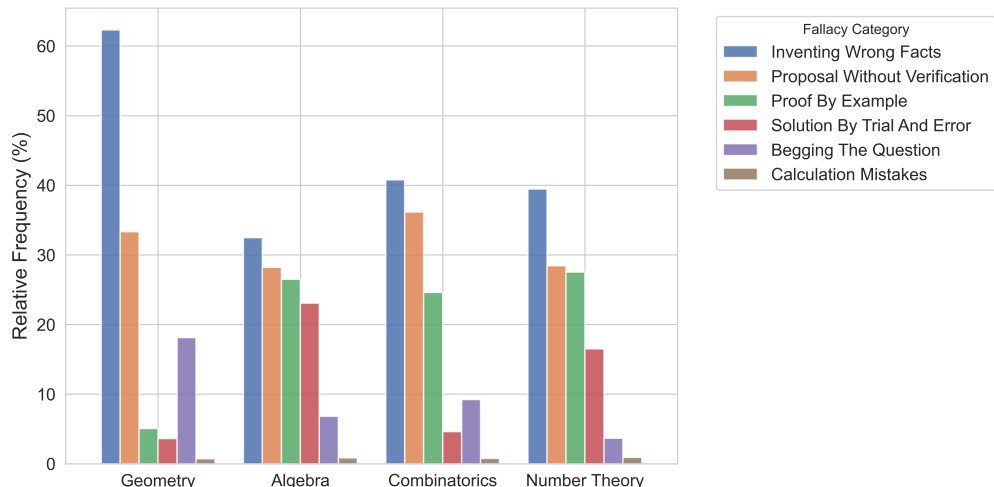

Figure 4: Relative frequencies of each fallacy in LLM-generated solutions among different topics

| Model | Real Solutions Correct (%) | Wrong Solutions Correct (%) |
|---|---|---|
| DeepSeek | 48 | 43 |
| Gemini 2.0 | 52 | 50 |
| o1 | 31 | 39 |
| o1-mini | 36 | 45 |
| o3-mini | 26 | 31 |

Table 3: Percentage of correct and incorrect solutions identified as *correct* by different LLMs during verification. The results illustrate the LLMs' difficulty in accurately distinguishing genuinely correct solutions from clearly incorrect ones containing explicit fallacies.

candidate solutions, and a reward model evaluates these candidates. The reward model can either be hard-coded (e.g., checking only the correctness of final answers), a learned reward model trained specifically for evaluation, or another powerful LLM serving as a judge (Plaat et al., 2024). A pertinent question in this context is whether state-of-the-art models can reliably distinguish fallacious generated solutions from correct, authentic solutions. We approach this question through two complementary analyses:

1. Do LLMs identify correct, authentic solutions as valid more frequently than fallacious generated solutions?

2. When presented with pairs consisting of a correct solution and an incorrect, fallacious one for each problem, are LLMs capable of accurately choosing the correct solution?

To investigate the first question, we gathered all problems from our evaluation dataset that had incorrect solutions generated by LLMs. Corresponding correct solutions for these problems were obtained from the Art of Problem Solving (AoPS) website[2]. We then prompted the LLMs to analyze each solution and explicitly request a final judgment of either *correct* or *wrong*. While this verification task can generally be nuanced due to solutions that partially satisfy correctness criteria, our selected examples distinctly represent either clearly incorrect solutions with obvious fallacies or entirely correct solutions. The description of all prompts we used in this section can be found in the appendix.

---

[2]https://artofproblemsolving.com/community/c3223_imo_shortlist

As shown in Table 3, DeepSeek and Gemini 2.0 produce responses of *correct* and *wrong* with nearly equal frequency for both genuinely correct solutions and incorrect LLM-generated solutions. Interestingly, the likelihood of identifying a genuinely correct solution as *correct* is even lower for the o1, o1-mini, and o3-mini models. These results demonstrate that the models are not suitable for use as judges, as they cannot reliably distinguish genuine solutions from obviously incorrect ones.

To investigate the second question, we applied a similar methodology. For each problem, we presented the models with pairs consisting of a correct solution and an incorrect, LLM-generated solution. We then prompted the models to identify the correct solution after careful analysis, explicitly informing them that exactly one solution was correct and the other was incorrect.

| Model | Accuracy (%) |
| --- | --- |
| DeepSeek | 57 |
| Gemini 2.0 | 49 |
| o1 | 50 |
| o1-mini | 46 |
| o3-mini | 52 |

Table 4: Accuracy of various LLMs in identifying the correct solution when presented with pairs consisting of one correct solution and one incorrect solution generated by an LLM.

As presented in Table 4, models o1, o1-mini, and Gemini 2.0 perform at or below random in distinguishing correct from incorrect solutions. Only DeepSeek and o3-mini perform modestly better than chance, outperforming random selection by 7% and 2%, respectively. These results indicate that the evaluated models currently have limited effectiveness as verifiers for challenging tasks such as IMO Shortlist-level problems.

## 6   General Insights into Gemini 2.5 Pro Solutions

Our evaluation of frontier LLMs on Olympiad-level mathematics revealed significant shortcomings in their ability to produce logically rigorous proofs and engage in genuine mathematical reasoning. Models such as OpenAI's o1, o1-mini, o3-mini, DeepSeek R1, indicate a reliance on heuristic shortcuts rather than authentic reasoning processes. Additionally, these LLMs demonstrated limited capability in effectively verifying solutions, performing at or near random levels when distinguishing correct proofs from clearly incorrect ones.

These findings emphasize two critical areas for improvement. First, there is a clear need for developing more sophisticated benchmarking methods that evaluate logical rigor and reasoning quality rather than merely assessing final answers. Second, relying solely on final answer correctness or utilizing more powerful LLMs as judges is insufficient; improved training schemas specifically designed to address the logical rigor of the proofs are essential for advancing future models toward human-level mathematical proficiency.

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

## A Human Evaluation Results with Gemini 2.5 Pro

We repeated the same process for 31 problems from the IMO shortlist 2014 problems with Gemini 2.5 Pro (preview-03-25). As shown in Table 5, Gemini 2.5 Pro significantly outperforms all other models. Qualitatively, Gemini 2.5 Pro solutions are noticeably more coherent and contain fewer basic errors. Additionally, Gemini 2.5 Pro generated a higher percentage of partially correct solutions. Even when solutions were incomplete, they often contained useful derivations and relevant proofs. In contrast, solutions from other models typically lacked useful information or meaningful insights.

| LLM Solution Model | Correct | Partially Correct | Incorrect |
|---|---|---|---|
| DeepSeek | 3.8% | 6.7% | 89.4% |
| Gemini 2.0 | 0.0% | 1.1% | 98.9% |
| **Gemini 2.5 Pro** | **25.8%** | **25.8%** | **48.4%** |
| o1 | 1.9% | 3.9% | 94.2% |
| o1-mini | 0.0% | 0.0% | 100.0% |
| o3-mini | 3.3% | 4.4% | 92.2% |

Table 5: Performance of different models on IMO shortlist problems (%)

Table 6 indicates that Gemini 2.5 Pro is more likely to provide a correct overall solution when its final answer is correct.

| Model | Final Answer Accuracy (%) | Correct\|Correct Final Answer (%) |
|---|---|---|
| DeepSeek | 63.2% | 0% |
| Gemini 2.0 | 43.8% | 0% |
| **Gemini 2.5** | 55.6% | **16.7%** |
| o1 | 30.8% | 12.5% |
| o1-mini | 35.0% | 0% |
| o3-mini | 48.3% | 14.3% |

Table 6: Comparison of evaluated LLMs highlighting the gap between final answer correctness and overall solution quality. **Final Answer Accuracy** denotes the percentage of correct final answers, whereas **Correct|Correct Final Answer** represents the percentage of fully correct solutions among instances where the final answer is correct.

Figure 5 shows the relative frequencies of various fallacies for each model, including Gemini 2.5 Pro. Gemini 2.5 notably makes fewer basic errors such as "Proposal Without Verification" and "Proof by Example," which typically have predictable patterns. Even when employing "Solution by Trial and Error," Gemini 2.5 generally attempts to check the other possible solutions and rule out alternatives. Interestingly, Gemini 2.5 introduces a unique pattern when it does "Inventing Wrong Facts". This occurs when the model initially attempts legitimate ideas and derivations but resorts to citing a non-existent theorem or result equivalent to the problem statement after failing. This behavior was common in nearly all challenging problems. We suspect that the root cause of this error is agent-based training. Since the model didn't have internet access during the evaluation, it hallucinated non-existent papers, blog posts, and theorems.

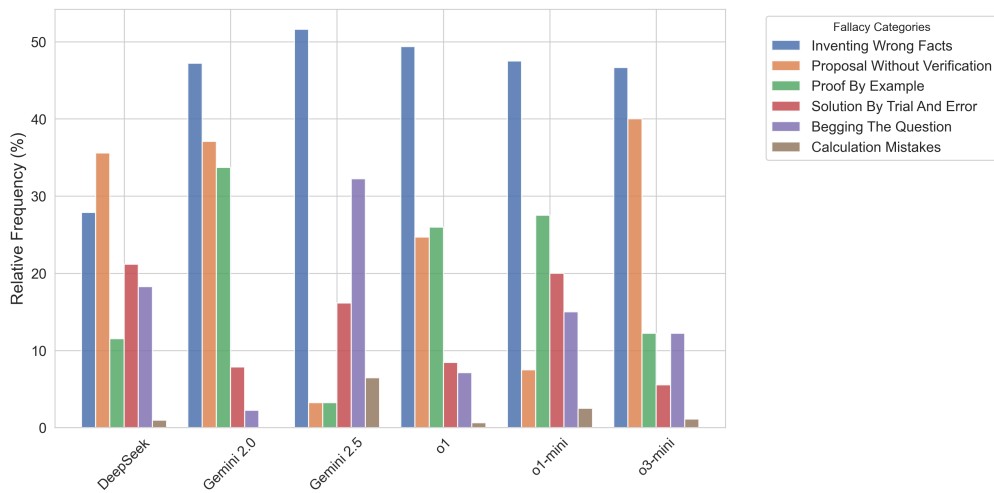

Figure 5: Relative frequencies of each fallacy for Gemini 2.5

Figures 6 and 7 illustrate the relative frequencies of fallacies categorized by problems with or without a final answer, and by topic, respectively. The previously mentioned patterns continue to hold for Gemini 2.5.

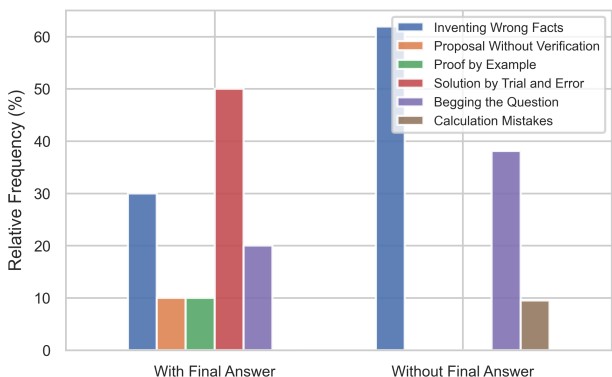

Figure 6: Relative frequencies of each fallacy in Gemini 2.5 solutions, comparing questions with and without a final answer.

## B  Automatic Evaluation Results

Table 7 shows each model's ability to classify genuine correct solutions versus LLM-generated incorrect solutions. Ideally, models should classify nearly 100% of correct solutions as correct and close to 0% of incorrect solutions as correct. Gemini 2.5 achieves 39% accuracy on correct solutions and 4% on incorrect solutions, resulting in the greatest distinction among all models. Other models perform near random chance.

Table 8 summarizes model accuracy in distinguishing correct solutions from incorrect solutions when presented together. Gemini 2.5 significantly outperforms other models with 83% accuracy, while others show near-random performance.

Notably, when Gemini 2.5 evaluated solutions generated by itself, its accuracy dropped to **60%** from 83%.

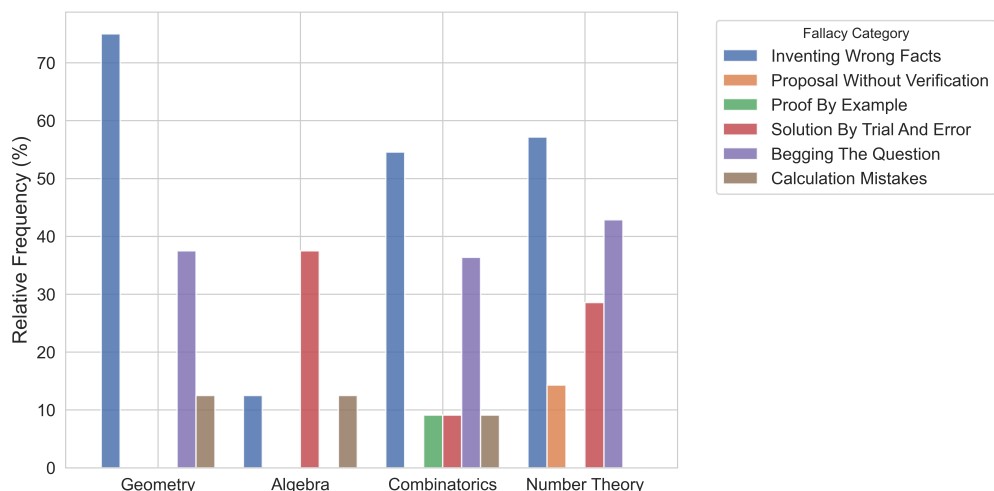

Figure 7: Relative frequencies of each fallacy in Gemini 2.5 solutions among different topics

| Model | Real Solutions Correct (%) | Wrong Solutions Correct (%) |
|---|---|---|
| DeepSeek | 48 | 43 |
| Gemini 2.0 | 52 | 50 |
| **Gemini 2.5** | **39** | **4** |
| o1 | 31 | 39 |
| o1-mini | 36 | 45 |
| o3-mini | 26 | 31 |

Table 7: Percentage of correct and incorrect solutions identified as *correct* by different LLMs during verification. The results illustrate the LLMs' difficulty in accurately distinguishing genuinely correct solutions from clearly incorrect ones containing explicit fallacies.

## C   Conclusion

We consistently observed better solutions from Gemini 2.5 compared to other reasoning models. Gemini 2.5 shows a clearer understanding of what a mathematical proof is, while other models mainly focus on finding equations or final answers without properly proving them. Because of this, other models often produce incoherent or unclear statements. This is why fallacies like "Proposal Without Verification" and "Proof by Example" happen less often with Gemini 2.5. Even when Gemini 2.5 uses "Solution by Trial and Error," it generally tries to rule out other possible solutions, although sometimes it doesn't fully check all possibilities.

| Model | Accuracy (%) |
|---|---|
| DeepSeek | 57 |
| Gemini 2.0 | 49 |
| Gemini 2.5 | 83 |
| o1 | 50 |
| o1-mini | 46 |
| o3-mini | 52 |

Table 8: Accuracy of various LLMs in identifying the correct solution when presented with pairs consisting of one correct solution and one incorrect solution generated by an LLM.

Geometry problems remain challenging for Gemini 2.5. The model either attempts to use complex coordinates, resulting in lengthy and unclear derivations, or uses geometric reasoning that still contains basic errors, similar to earlier models. Particularly, Gemini 2.5's geometry solutions often include trivially wrong claims or tautological facts. Despite these weaknesses, Gemini 2.5 significantly improves upon other reasoning models currently available.

## D   Concrete Examples for Common LLM Fallacies

### D.1   Illustrative Examples for Common LLM Fallacies

In this section, we present illustrative examples and comprehensive explanations for each fallacy, facilitating clearer interpretation and deeper conceptual understanding.

---

**Example: Proof by Example**

**Problem:** Prove that $3^{2n} - 1$ is divisible by 8 for all integers $n \geq 1$.

---

**Fallacious Solution:** To "prove" this statement, we can test the initial cases:
For $n = 1$,
$$3^2 - 1 = 8, \quad \text{which is divisible by 8.}$$
For $n = 2$,
$$3^4 - 1 = 81 - 1 = 80, \quad \text{which is divisible by 8.}$$
For $n = 3$,
$$3^6 - 1 = 729 - 1 = 728, \quad \text{which is divisible by 8.}$$
Hence, we "proved" that $3^{2n} - 1$ is divisible by 8 for all integers $n \geq 1$.

---

This argument is not a valid proof by induction, as it lacks the necessary inductive step. Without this inductive step, verifying a few initial cases does not guarantee the statement is true for all $n$. While the statement itself happens to be true in this example, the method of proof is fallacious.

---

**Example: Proposal Without Verification:**

**Problem:** Two players, Alice and Bob, take turns choosing a number from the set $\{1, 2, \ldots, n\}$, such that the sum of the chosen numbers never exceeds $2n$. The player unable to make a valid move loses. Prove that for all $n \geq 1$, there exists a winning strategy for one of the players.

---

**Fallacious Solution:** Alice always has a winning strategy. She should begin by picking the largest available number, and on each subsequent turn, she should pick numbers that ensure Bob cannot reach the limit of $2n$. By doing this, she can control the game and guarantee her victory. This approach is clearly optimal because by keeping Bob under pressure, Alice can always force him into a losing position."

---

While this strategy seems intuitive, the proposer does not provide any proof that it guarantees a win for Alice in all cases. For example, there is no justification that Alice's strategy prevents Bob from achieving the same goal or that Bob has no counter-strategy. Furthermore, no rigorous argument is provided to show that this method works for all values of $n$. A proper solution would involve analyzing the game mathematically, determining conditions under which Alice or Bob has a winning strategy, and proving these results using tools like

invariants or backward induction. Without this reasoning, the proposal remains incomplete and exemplifies the fallacy.

---

**Example: Inventing Wrong Facts**

**Problem:** Prove that the harmonic series

$$\sum_{n=1}^{\infty} \frac{1}{n}$$

diverges.

---

**Fallacious Solution:** By the *Harmonic Square Root Theorem*, the partial sum of the harmonic series up to $n$ terms is given by:

$$H_n = 2\sqrt{n} + \gamma + \mathcal{O}\left(\frac{1}{\sqrt{n}}\right),$$

where $\gamma$ is the Euler-Mascheroni constant. As $n \to \infty$, the dominant term $2\sqrt{n}$ grows without bound, while the constant $\gamma$ and error term $\mathcal{O}\left(\frac{1}{\sqrt{n}}\right)$ become negligible. Thus, $\lim_{n\to\infty} H_n = \infty$, proving divergence of the harmonic series.

---

The *Harmonic Square Root Theorem* is non-existent. While $H_n \sim \ln(n) + \gamma$ is a valid asymptotic approximation, claiming $H_n \propto \sqrt{n}$ is blatantly false.

---

**Example: Begging the Question**

**Problem:** Prove that the number $\sqrt{2}$ is irrational.

---

**Fallacious Solution:** Assume $\sqrt{2}$ is irrational. By definition, an irrational number cannot be expressed as a fraction $\frac{a}{b}$, where $a$ and $b$ are integers with $b \neq 0$. Since $\sqrt{2}$ cannot be written as $\frac{a}{b}$, it follows that $\sqrt{2}$ is irrational.

---

The argument begins by assuming the conclusion "Assume $\sqrt{2}$ is irrational". It then uses the definition of irrationality to "prove" the same claim, creating a logical loop. The conclusion ("$\sqrt{2}$ is irrational") is restated verbatim as the premise, offering no actual proof.

---

**Example: Solution by Trial and Error**

**Problem:** Find all functions $f : \mathbb{R} \to \mathbb{R}$ such that
$$f(x^2 - y^2) = xf(x) - yf(y), \quad \forall x, y \in \mathbb{R}.$$

---

**Fallacious Solution:**
Let us try some simple forms of $f(x)$: Suppose $f(x) = 0$. Substituting into the functional equation:
$$f(x^2 - y^2) = 0 \quad \text{and} \quad xf(x) - yf(y) = 0.$$
This satisfies the equation, so $f(x) = 0$ is a solution.
Now suppose $f(x) = x$. Substituting $f(x) = x$ into the equation:
$$f(x^2 - y^2) = x^2 - y^2 \quad \text{and} \quad xf(x) - yf(y) = x^2 - y^2.$$

This also satisfies the equation, so $f(x) = x$ is another solution.
Finally, consider $f(x) = kx$ for some constant $k$. Substituting:

$$f(x^2 - y^2) = k(x^2 - y^2) \quad \text{and} \quad xf(x) - yf(y) = k(x^2 - y^2).$$

This works for any $k$. So the solutions are $f(x) = 0$ and $f(x) = kx$ for any constant $k$.

This solution seems to rely on educated guesses to reach the correct answer but doesn't explain why other functions fail to satisfy the given functional equation.

---

**Example: Calculation Mistakes**

**Problem:** Let $x, y, z > 0$ satisfy $x + y + z = 6$. Find the maximum value of $xyz$.

---

**Fallacious Solution:**
Using the *AM-GM inequality*, we know:

$$\frac{x + y + z}{3} \geq \sqrt[3]{xyz}.$$

Substituting $x + y + z = 6$, we get:

$$\frac{6}{3} \geq \sqrt[3]{xyz}.$$

Simplifying:

$$2 \geq \sqrt[3]{xyz}.$$

Now, squaring both sides (instead of cubing):

$$4 \geq xyz.$$

Thus, we conclude that $xyz \leq 4$.

---

The critical error lies in Step 4, where the inequality was squared instead of being cubed. Since the inequality involves a cube root $\sqrt[3]{xyz}$, the proper operation should have been cubing both sides: $(2)^3 = 8$. This mistake leads to the incorrect conclusion $xyz \leq 4$, whereas the correct bound is $xyz \leq 27$. The solution becomes invalid because squaring distorts the inequality, making the result overly restrictive.

### D.2 Real Examples for Common LLM Fallacies

**Example: Proof by Example**

**Problem:** On a flat plane in Camelot, King Arthur builds a labyrinth $\mathfrak{L}$ consisting of $n$ walls, each of which is an infinite straight line. No two walls are parallel, and no three walls have a common point. Merlin then paints one side of each wall entirely red and the other side entirely blue. At the intersection of two walls there are four corners: two diagonally opposite corners where a red side and a blue side meet, one corner where two red sides meet, and one corner where two blue sides meet. At each such intersection, there is a two-way door connecting the two diagonally opposite corners at which sides of different colours meet. After Merlin paints the walls, Morgana then places some knights in the labyrinth. The knights can walk through doors, but cannot walk through walls.
Let $k(\mathfrak{L})$ be the largest number $k$ such that, no matter how Merlin paints the labyrinth $\mathfrak{L}$, Morgana can always place at least $k$ knights such that no two of them can ever

meet. For each $n$, what are all possible values for $k(\mathfrak{L})$, where $\mathfrak{L}$ is a labyrinth with $n$ walls?

---

**Fallacious Solution:**
For each integer $n \geq 1$, the labyrinth $\mathfrak{L}$ constructed by King Arthur consists of $n$ non-parallel, infinitely extending walls with no three walls intersecting at a single point. Merlin paints each wall with one side red and the other blue. The intersections of these walls create doors that connect specific regions within the labyrinth.

**Understanding $k(\mathfrak{L})$:**
The value $k(\mathfrak{L})$ represents the largest number $k$ such that, regardless of how Merlin paints the labyrinth, Morgana can always place at least $k$ knights in the labyrinth where no two knights can meet through the doors.

**Analyzing Small Cases:**
**1. For $n = 1$:**
- There is only one wall, dividing the plane into two regions.

- Since there are no intersections, there are no doors.

- Morgana can place one knight in each region, so $k(\mathfrak{L}) = 2$.

**2. For $n = 2$:**
- Two intersecting walls create four regions.

- There is one door connecting two diagonally opposite regions.

- Morgana can place three knights such that no two are connected by the door, so $k(\mathfrak{L}) = 3$.

**3. For $n = 3$:**
- Three walls intersecting pairwise create seven regions.

- There are three doors, each connecting two regions.

- Morgana can ensure that four knights are placed without any two being connected by a door, so $k(\mathfrak{L}) = 4$.

**Generalizing the Pattern:**
Observing the pattern from the small cases, we see that:
- **For each $n$, $k(\mathfrak{L}) = n + 1$.**

This means that no matter how Merlin paints the labyrinth, Morgana can always place $n + 1$ knights such that no two knights can meet through the doors. This result stems from the combinatorial structure of the labyrinth and the way doors are created at intersections.

**Conclusion:**
For any number of walls $n$, the largest guaranteed number of knights $k(\mathfrak{L})$ that Morgana can place without any two being able to meet is always $n + 1$.

**Answer:**
For every number $n$ of walls, the maximal $k$ is $n + 1$. Thus, each $k(\mathfrak{L})$ equals $n + 1$.

---

The proof claims that $k(\mathfrak{L}) = n + 1$ by testing it for small $n$s and then it generalizes this claim for all natural $n$s without justification.

---

**Example: Proposal Without Verification**

**Problem:** A social network has 2019 users, some pairs of whom are friends. Whenever user $A$ is friends with user $B$, user $B$ is also friends with user $A$. Events of the following kind may happen repeatedly, one at a time: Three users $A$, $B$, and $C$ such that $A$ is friends with both $B$ and $C$, but $B$ and $C$ are not friends, change their

---

friendship statuses such that $B$ and $C$ are now friends, but $A$ is no longer friends with $B$, and no longer friends with $C$. All other friendship statuses are unchanged. Initially, 1010 users have 1009 friends each, and 1009 users have 1010 friends each. Prove that there exists a sequence of such events after which each user is friends with at most one other user.

---

**Fallacious Solution:**
To solve this problem, we need to demonstrate that it's possible to transform the given social network into a state where each user is friends with at most one other user using the specified operations.

**Initial Setup**
- **Number of Users:** 2019

- **Friendship Structure:**

  - 1010 users have 1009 friends each.
  - 1009 users have 1010 friends each.

- **Total Friendships Initially:**

$$1010 \times 1009 + 1009 \times 1010 = 2 \times 1010 \times 1009$$

  This counts each friendship twice (once for each user in the pair), so the actual number of unique friendships is:

$$\text{Total Edges} = 1010 \times 1009$$

**Objective**
Transform the network so that each user has **at most one friend**. In graph theory terms, we aim to reach a state where the graph is a **matching**, meaning it consists of disjoint edges with no overlaps.

**Operation Details**
The operation allows us to:

- **Select Three Users:** $A, B, C$ such that:

  - $A$ is friends with both $B$ and $C$.
  - $B$ and $C$ are **not** friends.

- **Modify Friendships:**

  - **Add:** Friendship between $B$ and $C$.
  - **Remove:** Friendships between $A$ and $B$, and between $A$ and $C$.

**Effect of Each Operation**
- **Total Friendships Decrease:** Each operation removes two friendships and adds one, resulting in a net decrease of one friendship.

- **Degree Changes:**

  - $A$'s degree decreases by 2.
  - $B$ and $C$'s degrees remain unchanged since one friendship is removed and another is added.

**Strategy to Achieve the Objective**
- **Reducing Degrees:**

  - Repeatedly apply the operation to reduce the degrees of users with more than one friend.
  - Each operation ensures that a user's degree decreases, moving towards the desired state.

- **Termination Condition:**

  - The process continues until no user has more than one friend.
  - If a user still has more than one friend, the operation can still be applied because there will always be at least two friends who aren't connected (given the initial high degree and the specific structure of operations).

- **Final State:**

  - The network will consist of disjoint pairs of friends (a matching) and possibly some users with no friends.
  - This ensures that no user has more than one friend.

**Conclusion**

Through systematic application of the specified operation, we can decrementally reduce the degrees of all users with more than one friend. Given the initial conditions and the nature of the operation, it's guaranteed that we can reach a state where each user has at most one friend.

**Final Answer:** After performing suitable operations, the network can be transformed so no user has more than one friend. Thus, such a sequence exists.

As we can see, the model has proposed a sequence but doesn't do any mathematical verification of its correctness.

---

**Example: Inventing Wrong Facts**

**Problem:** You are given a set of $n$ blocks, each weighing at least 1; their total weight is $2n$. Prove that for every real number $r$ with $0 \leq r \leq 2n - 2$ you can choose a subset of the blocks whose total weight is at least $r$ but at most $r + 2$

---

**Fallacious Solution:**

To prove that for any real number $r$ with $0 \leq r \leq 2n - 2$, there exists a subset of blocks whose total weight is at least $r$ but at most $r + 2$, we can proceed as follows:

**Key Observations:**

1. **Block Weights and Total Weight:**

   - Each block has a weight of at least 1.
   - The total weight of all $n$ blocks is $2n$.

2. **Subset Sums:**

   - The possible subset sums range from 0 (empty set) to $2n$ (all blocks).
   - Since each block has a weight of at least 1, any two distinct subset sums differ by at least 1.

3. **Covering the Interval** $[0, 2n - 2]$**:**

   - We need to ensure that for any $r$ in $[0, 2n - 2]$, there exists a subset sum $s$ such that $r \leq s \leq r + 2$.

**Proof Strategy:**

**Sequential Coverage**

- List all possible subset sums in increasing order: $s_0 = 0, s_1, s_2, \ldots, s_m = 2n$.

- Each subsequent sum satisfies $s_{i+1} - s_i \geq 1$ because each block has a weight of at least 1.

**Interval Overlap**

- Consider intervals of the form $[s_i, s_i + 2]$ for each subset sum $s_i$.

- Since $s_{i+1} \geq s_i + 1$, the next interval $[s_{i+1}, s_{i+1} + 2]$ starts no more than 1 unit after the start of the previous interval.

- Therefore, the union of these intervals $[s_i, s_i + 2]$ covers the entire range $[0, 2n]$ without gaps larger than 2.

**Ensuring Coverage for Any $r$**

- For any $r$ in $[0, 2n - 2]$, there exists some subset sum $s_i$ such that $s_i \leq r \leq s_i + 2$.

- This guarantees that $r$ lies within the interval $[s_i, s_i + 2]$, meaning there exists a subset with a sum in $[r, r + 2]$.

**Conclusion** By sequentially covering the range $[0, 2n]$ with overlapping intervals of length 2 derived from the subset sums, we ensure that for any $r$ within the specified range, there exists a suitable subset whose total weight satisfies $r \leq \text{sum}(S) \leq r + 2$.
**Answer** By ordering all subset sums and noting that each differs by at least 1, the intervals $[s, s + 2]$ cover $[0, 2n - 2]$. Thus, for any $r$ between 0 and $2n - 2$, some subset has total weight at least $r$ and at most $r + 2$.

The model invents a wrong fact/observation and builds its argument around it. The main fact that is used for the proof is obviously not true. Consider $\{1, 1.5, 2, 3.5\}$.

---

**Example: Begging the Question**

**Problem:** A polynomial $P(x, y, z)$ in three variables with real coefficients satisfies the identities

$$P(x, y, z) = P(x, y, xy - z) = P(x, zx - y, z) = P(yz - x, y, z).$$

Prove that there exists a polynomial $F(t)$ in one variable such that

$$P(x, y, z) = F(x^2 + y^2 + z^2 - xyz).$$

---

**Fallacious Solution:**
To prove that there exists a polynomial $F(t)$ in one variable such that

$$P(x, y, z) = F(x^2 + y^2 + z^2 - xyz),$$

we analyze the given conditions and the structure of the polynomial $P(x, y, z)$.
**Step 1: Define the Invariant** Let's define the invariant:

$$t = x^2 + y^2 + z^2 - xyz.$$

Our goal is to show that $t$ remains unchanged under the transformations provided in the problem statement.
**Step 2: Verify Invariance Under Given Transformations** We need to verify that $t$ is invariant under the following transformations:
**1. First Transformation**
$$P(x, y, z) = P(x, y, xy - z).$$
Compute $t(x, y, xy - z)$:

$$t = x^2 + y^2 + (xy - z)^2 - xy(xy - z).$$

Expanding this:

$$t = x^2 + y^2 + x^2y^2 - 2xyz + z^2 - x^2y^2 + xyz = x^2 + y^2 + z^2 - xyz.$$

Thus, $t$ remains unchanged.

**2. Second Transformation**

$$P(x, y, z) = P(x, zx - y, z).$$

Compute $t(x, zx - y, z)$:

$$t = x^2 + (zx - y)^2 + z^2 - x(zx - y)z.$$

Expanding this:

$$t = x^2 + z^2x^2 - 2xyz + y^2 + z^2 - x^2z^2 + xyz = x^2 + y^2 + z^2 - xyz.$$

Again, $t$ remains unchanged.

**3. Third Transformation**

$$P(x, y, z) = P(yz - x, y, z).$$

Compute $t(yz - x, y, z)$:

$$t = (yz - x)^2 + y^2 + z^2 - (yz - x)yz.$$

Expanding this:

$$t = y^2z^2 - 2xyz + x^2 + y^2 + z^2 - y^2z^2 + xyz = x^2 + y^2 + z^2 - xyz.$$

Once more, $t$ remains unchanged.

**Step 3: Conclude the Existence of $F(t)$** Since $t = x^2 + y^2 + z^2 - xyz$ is invariant under all the given transformations, and $P(x, y, z)$ remains unchanged under these transformations, it follows that $P$ depends only on $t$.
Therefore, there exists a polynomial $F(t)$ such that:

$$P(x, y, z) = F(t) = F(x^2 + y^2 + z^2 - xyz).$$

**Conclusion** Any polynomial $P(x, y, z)$ satisfying the given identities must be a polynomial in the single variable $t = x^2 + y^2 + z^2 - xyz$.
**Answer** Because $x^2 + y^2 + z^2 - xyz$ is unchanged by the given equalities, $P$ depends only on it. Thus, there is an $F$ with:

$$P(x, y, z) = F(x^2 + y^2 + z^2 - xyz).$$

The generated solution begs the question. The fact $P(x, y, z) = F(x^2 + y^2 + z^2 - xyz)$. is the problem statement itself, and it needs to be proven.

**Example: Solution by Trial and Error**

**Problem:** Let $\mathbb{Z}$ be the set of integers. Determine all functions $f : \mathbb{Z} \to \mathbb{Z}$ such that, for all integers $a$ and $b$,

$$f(2a) + 2f(b) = f(f(a + b)).$$

Proposed by Liam Baker, South Africa

---

**Fallacious Solution:**
To determine all functions $f : \mathbb{Z} \to \mathbb{Z}$ that satisfy the functional equation

$$f(2a) + 2f(b) = f(f(a + b)) \quad \text{for all integers } a \text{ and } b,$$

let's explore possible forms of $f$.
**Step 1: Assume a Linear Function**

First, consider that $f$ might be a linear function of the form:

$$f(n) = kn + c,$$

where $k$ and $c$ are constants in $\mathbb{Z}$.
Substituting into the functional equation:

$$f(2a) + 2f(b) = k(2a) + c + 2(kb + c) = 2ka + c + 2kb + 2c = 2k(a + b) + 3c.$$

On the other hand:

$$f(f(a + b)) = f(k(a + b) + c) = k(k(a + b) + c) + c = k^2(a + b) + kc + c.$$

Setting the two expressions equal gives:

$$2k(a + b) + 3c = k^2(a + b) + kc + c.$$

For this to hold for all integers $a$ and $b$, the coefficients must match:

$$2k = k^2 \quad \text{and} \quad 3c = kc + c.$$

Solving the first equation:

$$k^2 - 2k = 0 \implies k(k - 2) = 0 \implies k = 0 \text{ or } k = 2.$$

**Case 1:** $k = 0$

$$3c = c \implies 2c = 0 \implies c = 0.$$

So, $f(n) = 0$ is a solution.

**Case 2:** $k = 2$

$$3c = 3c \quad \text{(always true).}$$

Here, $c$ can be any integer. Thus, $f(n) = 2n + c$ is a solution for any integer $c$.
**Step 2: Verify Non-Linear Solutions** Assuming $f$ is not linear leads to more complex conditions that are difficult to satisfy uniformly across all integers. Hence, no non-linear solutions appear to exist beyond the linear forms already identified.
**Conclusion** All solutions to the functional equation are either the constant zero function or affine linear functions of the form $f(n) = 2n + c$, where $c$ is an integer constant.
**Final Answer:** All integer functions equal to twice $n$ plus a fixed constant. In other words, every $f$ with $f(n) = 2n + c$ for some integer $c$.

The generated proof only tests the problem's equation for a special class and concludes the result. This is an example of a solution by trial and error fallacy.

### Example: Calculation Mistakes

**Problem:** Determine all integers $n \geq 2$ having the following property: for any integers $a_1, a_2, \ldots, a_n$ whose sum is not divisible by $n$, there exists an index $1 \leq i \leq n$ such that none of the numbers

$$a_i, a_i + a_{i+1}, \ldots, a_i + a_{i+1} + \ldots + a_{i+n-1}$$

is divisible by $n$. Here, we let $a_i = a_{i-n}$ when $i > n$.

---

**Fallacious Solution:**
**We will show that among the integers $n \geq 2$, the property**
*"For any integers $a_1, a_2, \ldots, a_n$ whose sum is not divisible by $n$, there is some index $1 \leq i \leq n$ so that none of the numbers*

$$a_i, \ a_i + a_{i+1}, \ \ldots, \ a_i + a_{i+1} + \cdots + a_{i+n-1}$$

*is divisible by $n$"*
holds if and only if $n = 2$.
In other words, if $n \geq 3$ then one may construct a counterexample sequence, while for $n = 2$ the property always holds.

In what follows, we describe a method to translate the condition into one involving a cyclic sequence of residues and then show that counterexamples exist for every $n \geq 3$, but not for $n = 2$.

**Reformulating the Problem**

For a given sequence $a_1, \ldots, a_n$, define the cumulative sums

$$S_k = a_1 + a_2 + \cdots + a_k \quad \text{for } k = 1, \ldots, n, \qquad S_0 = 0.$$

All congruences below are modulo $n$.

Note that for any index $i$ (with indices taken cyclically modulo $n$, i.e., $a_{j+n} = a_j$), the partial sums

$$a_i, \; a_i + a_{i+1}, \; \ldots, \; a_i + a_{i+1} + \cdots + a_{i+n-1}$$

can be written as

$$S_i - S_{i-1}, \; S_{i+1} - S_{i-1}, \; \ldots, \; S_{i+n-1} - S_{i-1}.$$

Thus, none of these numbers is divisible by $n$ if and only if

$$S_{i-1} \not\equiv S_{i-1+j} \quad \text{for } j = 1, 2, \ldots, n.$$

In other words, in the list $S_0, S_1, \ldots, S_n$ (indices modulo $n+1$), we seek an index $t$ (with $t = i - 1$) such that the residue $S_t$ is isolated—that is, no other $S_k$ is congruent to $S_t$ modulo $n$.

Since $\mathbb{Z}/n\mathbb{Z}$ has $n$ elements, but we are choosing $n+1$ values $S_0, \ldots, S_n$, the pigeonhole principle guarantees that at least one residue is repeated. However, for the property to hold, we only need that at least one $S_t$ be unique among the $S_k$.

Thus, the problem reduces to the following:

> *For which integers $n \geq 2$ is it true that for every choice of integers $a_1, \ldots, a_n$ with $S_n \not\equiv 0$ there exists $t \in \{0, 1, \ldots, n\}$ such that*
>
> $$S_t \not\equiv S_k \quad \text{for all } k \neq t ?$$

If such a $t$ always exists, then setting $i = t + 1$ ensures that all partial sums starting at $a_i$ are nonzero modulo $n$.

**Counterexamples for $n \geq 3$**

We now describe how to construct counterexamples when $n \geq 3$.

- It is possible to construct a sequence $S_0, S_1, \ldots, S_n$ in $\mathbb{Z}/n\mathbb{Z}$, with $S_0 = 0$ and $S_n \not\equiv 0$, such that every residue appears at least twice.

- In such a configuration, no $S_t$ is unique, so the desired property fails.

One simple way to construct such a sequence is to alternate between two values (e.g., 0 and $x$), for example:

$$S_0 = 0, \quad S_1 = 0, \quad S_2 = x, \quad S_3 = 0, \quad S_4 = x, \quad \ldots, \quad S_n = x$$

where $x \not\equiv 0 \mod n$. Then define

$$a_k = S_k - S_{k-1}, \quad \text{for } k = 1, \ldots, n.$$

This ensures that $a_1 + \cdots + a_n = S_n \not\equiv 0$, and because every residue among $S_0, \ldots, S_n$ appears at least twice, the property fails.

Thus, for every $n \geq 3$, one can construct a sequence where no index $i$ satisfies the given condition.

**The Case $n = 2$**

Now consider $n = 2$. In $\mathbb{Z}/2\mathbb{Z}$, there are only two residues: 0 and 1.

- The sequence $S_0, S_1, S_2$ has 3 elements in a 2-element set.

- By the pigeonhole principle, at least one residue appears twice.

- Therefore, the third must appear only once. So there exists a unique $S_t$.

Hence, the property *must* hold for $n = 2$.
A concrete check:

- If $a_1$ is odd and $a_2$ is even, then $a_1 \equiv 1$, $a_1 + a_2 \equiv 1$.

- If $a_1$ is even and $a_2$ is odd, then taking $i = 2$, $a_2 \equiv 1$, and $a_2 + a_1 \equiv 1$.

So in both cases, the two partial sums are nonzero modulo 2, confirming the property.
**Conclusion**

- For all $n \geq 3$, one may construct a sequence $a_1, \ldots, a_n$ such that for every $i$, at least one of the cyclic partial sums is divisible by $n$.

- For $n = 2$, such a counterexample is impossible, and the property always holds.

Therefore, the only integer $n \geq 2$ for which the property holds is:

$$\boxed{2}.$$

**Final Answer:** The property holds if and only if $n = 2$.

If $n$ is odd, then $S_n$ will be 0, which means the sum of $a_1 + a_2 + \cdots + a_n$ is divisible by $n$, and this contradicts the problem statement. The model has made a calculation mistake in this case and failed to realize that $S_n \neq x$ for odd $n$s

### D.3 Evaluation Prompts

---

**Prompt: Solution Evaluation Prompt**

You are an expert mathematician tasked with rigorously evaluating the correctness of a proposed solution to a mathematical problem, specifically in the style of the International Mathematical Olympiad (IMO).

Below, you will receive:
1. **The Problem Statement:** A description of the problem to be solved.

2. **The Candidate Solution:** A step-by-step explanation of a proposed solution to the problem.

Your evaluation should include:

- **Logical Analysis:** Carefully check each step for logical consistency, correctness of mathematical reasoning, and appropriateness of methods used.

- **Identification of Errors:** Clearly pinpoint and describe any mathematical errors, flawed logic, unjustified assumptions, or missing steps.

- **Clarification:** Briefly explain why each identified issue is problematic, referring explicitly to mathematical principles or concepts involved.

Provide your evaluation strictly adhering to the following structured response format:

---

**Analysis:**
{Provide a thorough, step-by-step analysis of the candidate solution. Clearly indicate if and where logical fallacies, incorrect assumptions, or mathematical inaccuracies occur. If the solution is fully correct, justify your reasoning thoroughly.}

**Final Answer:**
Clearly indicate the result of your analysis using exactly one of the following formats:

$\boxed{\text{Correct}}$ or $\boxed{\text{Wrong}}$

---

**Prompt: Solution Comparison Prompt**

I will present a problem along with two candidate solutions. Your task is to evaluate the solutions and determine which one is correct. Note that **one solution is always correct and one solution is always wrong**.

The primary criterion for your judgment is the **correctness of the arguments** provided in each solution.

Please respond using the following template:

---

**Analysis:**
{Provide your analysis of the solutions here, discussing the strengths and weaknesses of their arguments.}

**Final Answer:**

{ $\boxed{\text{Solution 1}}$ or $\boxed{\text{Solution 2}}$ }

## D.4 Additional Notes

To mitigate recency bias, the solutions provided by the LLM and the real solutions were randomly permuted during the comparative analysis experiment. Consistent outcomes were observed across various prompts, indicating robustness in the experimental setup. Furthermore, our investigation revealed that automated rating rubrics, such as the LLM-as-a-judge approach, yielded similar comparative results. Notably, frontier LLMs did not assign significantly higher ratings to correct solutions compared to incorrect ones. Due to constraints in space, the present discussion is intentionally restricted to the binary evaluation scenario.

