# OpenReview forum: "Brains vs. Bytes: Evaluating LLM Proficiency in Olympiad Mathematics"
_colmweb.org/COLM/2025/Conference — COLM 2025_

### Official Review · Reviewer_ZMU4 · 2025-04-28

**Rating:** 8
**Confidence:** 4
**Ethics Flag:** 1

**Summary:**

The paper critically examines the ability of large language models (LLMs) to solve challenging mathematical problems from the International Mathematics Olympiad (IMO). The authors conduct qualitative and quantitative evaluations of LLM-generated proofs, revealing that current LLMs struggle with Olympiad-level problems and often fail to distinguish between correct and flawed mathematical reasoning. The study emphasizes that LLMs often rely on pattern recognition and heuristic shortcuts rather than genuine mathematical reasoning, highlighting a significant gap between LLM performance and human expertise in advanced mathematical problem-solving.

The study design involves both qualitative and quantitative evaluations, which strengthens the validity of the findings. The methodology seems rigorous, with a focus on identifying and categorizing fallacies in LLM-generated solutions. The use of human evaluators with expertise in mathematics further enhances the reliability of the evaluation.

The introduction effectively sets the context by discussing recent advancements in LLMs and their performance on mathematical reasoning tasks. The authors clearly outline their methodology, including the problem selection rationale and the process of evaluating LLM-generated solutions.

The originality of this work lies in its focus on evaluating the logical rigor of LLM-generated proofs rather than solely relying on the correctness of final answers. The study addresses a significant gap in current evaluation benchmarks, which often overlook the importance of sound mathematical reasoning. The identification and categorization of common fallacies in LLM solutions also contribute to the originality of the research.

The significance of this study is highlighted by its potential impact on the development of more effective benchmarks for evaluating mathematical reasoning in LLMs. The findings underscore the limitations of current LLMs in solving challenging Olympiad-level problems and emphasize the need for training strategies that prioritize the validity and coherence of mathematical arguments. The study also provides insights into the feasibility of bootstrapping verification processes to improve the quality of LLM-generated proofs.

**Questions To Authors:**

1. Have you considered using the same rubric of fallacy recognition and analysis for existing olympiad benchmarks? If so, provide more details.

2. How do you plan to share the results? I did not see any of the code or prompt? Did you employ any prompting strategy?

**Reasons To Accept:**

1. The paper evaluates LLMs on their ability to produce logically sound mathematical proofs, rather than just focusing on the correctness of the final answer. This is significant because it addresses a critical gap in current benchmarks, which often fail to assess the quality of reasoning. The study systematically identifies and categorizes common logical fallacies in LLM-generated solutions. This provides valuable insights into the limitations of current LLMs and offers a framework for improving their mathematical reasoning abilities.

2. The research work involves human expert evaluation that exposes further insights into LLM behavior

3. The research investigates whether LLMs can verify the correctness of their own solutions, which could lead to the development of bootstrapping verification processes. This has the potential to enable LLMs to improve the quality of their proofs and reduce reliance on human evaluation.

**Reasons To Reject:**

LLMs lack of mathematical reasoning is well defined. It is interesting to see the improvement in details provided by experts. However, perhaps the contribution might be strengthened by the evaluation on existing benchmarks as well.

---

> ### Author Response · Authors · 2025-05-30
>
> We're truly grateful for the reviewer's insightful and encouraging evaluation, which affirms the significance and clarity of our work. We especially appreciate the praise for our original focus on logical rigor over just final answers, which addresses a critical gap in current benchmarks. Their recognition of our rigorous methodology, particularly the systematic fallacy categorization and the value of human expert evaluation, confirms the strength of our approach. We're also excited by the reviewer's insight into how our research could inform the development of improved training strategies for LLMs.
>
> ### **Addressing "Reasons to Reject"**
>
> **LLMs' Lack of Mathematical Reasoning is Well-Defined, But Evaluation on Existing Benchmarks is Missing:**
>
> The reviewer correctly notes that LLMs' lack of mathematical reasoning is a known area of research, and our paper provides further detailed insights from experts. We acknowledge the suggestion to evaluate on existing benchmarks. However, as discussed in our paper, many current benchmarks, such as GSM8K and MATH, largely consist of routine problems that have been widely circulated. Not only are the problems and their solutions repetitive, but even individual solution steps are often highly formulaic. This makes these benchmarks less suitable for measuring LLMs' capabilities in generating solutions that require **creativity and novel insights**, which is a core focus of our work with Olympiad-level problems. Our aim was to specifically probe the limits of LLMs beyond mere recall or mechanical application of common methods.
>
> ---
>
> ### Response to Questions to Authors
>
> **1\. Have you considered using the same rubric of fallacy recognition and analysis for existing Olympiad benchmarks?**
>
> The main issue with applying our rubric to other available Olympiad benchmarks (like OmniMath and OlympiadBench) is their primary focus on problems with a **concrete final answer**. A significant portion of our work, and a key strength, lies in evaluating **proof-based problems** which are not covered by the existing benchmarks. We chose IMO shortlist problems specifically for their originality, multi-step nature, and the creativity required for their solutions, qualities that are less consistently present in other lower-tier contests or benchmarks.
>
> **2\. How do you plan to share the results? I did not see any of the code or prompt? Did you employ any prompting strategy?**
>
> **Yes, we plan to release the annotated dataset publicly.** We are currently working on adding more comprehensive annotations and aim to make it available by the end of summer.
>
> Regarding our methodology, we employed a **simple one-step prompting strategy** for generating proofs. The specific prompts used for our automated analysis section are provided in the appendix of the paper. While we did not systematically investigate different prompting strategies, our manual inspections suggest that the choice of prompt does not significantly alter the fundamental findings. If a model lacks the intrinsic capability to generate rigorous proofs for Olympiad-level problems, minor variations in prompting strategies generally do not substantially improve the quality of the response.

---

> > ### Comment · Reviewer_ZMU4 · 2025-06-03
> >
> > Thanks for the clarifying comments.

---

### Official Review · Reviewer_hUEG · 2025-05-11

**Rating:** 7
**Confidence:** 4
**Ethics Flag:** 1

**Summary:**

The paper questions the performance of language models on difficult mathematical reasoning tests. They find that the explanations and reasoning chains produced by the language models often contain mistakes and fallacies, calling into question their true ability for mathematical reasoning

**Reasons To Accept:**

I find this type of work to be valuable to the community because it points out a possible overestimation of language model capabilities. I believe that the categorization is helpful for understanding how language models make mistakes.

**Reasons To Reject:**

One issue with the methodology is that just because the surface form of the chain of thought reasoning may contain incomplete proofs that does not mean that the model isn’t capable of generating complete logical proofs. Some of the reasoning may be taking place internally to the model and may not be surfaced during generation. Indeed the high performance of these models may indicate that such internal reasoning is happening.

That being said, the finding that models cannot distinguish correct from incorrect proofs does reinforce the hypothesis that the models are doing incorrect reasoning. The only alternative hypothesis I can think of is that the models have some preferences for their own generations, causing them to favor the incorrect proofs.

---

> ### Author Response · Authors · 2025-05-30
>
> We thank the reviewer for their positive feedback and for highlighting the value of our work.
> The reviewer raises an important point about our methodology: the possibility that logical reasoning might occur within the Chain-of-Thought (CoT) but fail to appear in the final proof.
>
> CoT outputs were not consistently available across all models we evaluated. Even when available, systematically analyzing raw CoTs is highly labor-intensive and challenging. CoTs are often significantly longer than the final response, lack clear structure, and frequently contain failed attempts or irrelevant detours, making rigorous, large-scale human evaluation impractical.
>
> That said, we disagree with the hypothesis that models may internally produce correct reasoning within their CoTs but fail to express it in the final answer. While we did not conduct a comprehensive human evaluation of all CoTs, we manually examined a subset in cases where CoTs were available, motivated by the same concern raised by the reviewer. In these cases, we found no evidence of complete, sound proofs within the CoT that were then omitted in the final output. Although some CoTs included additional calculations or exploratory reasoning, these did not amount to rigorous or correct arguments that were distinct from the flawed logic present in the final response.
>
> Furthermore, as the reviewer rightly notes, if models were indeed capable of correct internal reasoning, they should be able to distinguish between valid and invalid proofs. However, as shown in Section 5, models failed to reliably identify even blatantly incorrect proofs containing naive errors. If a model were capable of constructing a sound proof internally, we would expect it to succeed at such verification tasks. Its failure to do so strongly challenges the notion that correct reasoning is happening internally but simply not being communicated.
>
> In summary, we believe the errors observed in the final proofs reflect a deeper issue: a fundamental limitation in the models’ ability to perform rigorous reasoning, rather than a mere failure in surfacing internally correct thought processes.

---

> > ### Comment · Reviewer_hUEG · 2025-06-03
> >
> > I see that I had a slight misunderstanding of your paper, as I had assumed that you also evaluated chains of thought (where available). In turn, when I said "internal" chains of thought, I really meant internal *to the model*, i.e., in the latent space/hidden states. These are inherently harder to inspect and draw conclusions about, and I wouldn't expect you to---my point was that we cannot rule out the hypothesis that *internally* the model has well-formed/logically sound explanations that never get surfaced in the either the CoT or final output. The failure to identify flawed proofs does support your hypothesis, but does not fully rule out the possibility of internal rigor, because the model may prefer its own outputs regardless of correctness.
> >
> > That being said, this is a minor limitation/caveat in my view, and acknowledging (or not acknowledging it) has little bearing on the overall quality of the paper. I maintain my positive review and recommendation for acceptance.

---

### Official Review · Reviewer_qtx8 · 2025-05-12

**Rating:** 5
**Confidence:** 4
**Ethics Flag:** 1

**Summary:**

This paper investigates the proficiency of Large Language Models (LLMs) in solving Olympiad-level mathematical problems, emphasizing the logical rigour of proofs rather than just the correctness of final answers. The authors conducted qualitative and quantitative human evaluations of proofs and solutions generated by frontier LLMs, including OpenAI's o1, o1-mini, o3-mini, DeepSeek R1, and Gemini 2.0. The authors mention a schema for assessing the reasoning capabilities of LLMs and categorized common errors.

The study evaluates models on 455 IMO shortlist problems from 2009 to 2023, covering algebra, combinatorics, geometry, and number theory. The findings reveal that current LLMs fall short of solving challenging olympiad problems with a negligible percentage of solutions being fully correct or providing meaningful insights. The paper highlights that occasional correct final answer from LLMs often stem from pattern recognition or heuristic shortcuts rather than genuine math reasoning. Further, the LLMs struggle to distinguish between correct and flawed solutions, performing at or near random levels in verification. The paper mentions that there is a substantial gap between LLM performance and human expertise in advanced mathematical reasoning.

**Questions To Authors:**

Could you elaborate on the inter-annotator agreement for the classification of fallacies and solution correctness? How was consistency ensured among the seven evaluators, especially for borderline cases?

The paper mentions that "frontier models may have been trained on publicly available high-quality mathematical datasets". Given the observation that models often rely on heuristics and pattern recognition, did the evaluators consider cases where models might have memorized solutions to specific IMO problems or similar problem structures? How was this potential bias mitigated in the evaluation process?

For the "Proposal Without Verification" fallacy , the paper notes that models might struggle to determine what calculations and statements to include in the final response, sometimes producing vague claims like "It is easy to show that...". Are there any insights into why LLMs omit these essential steps or introduce incorrect statements instead of providing rigorous arguments?

Given that "Inventing Wrong Facts" was the most frequent fallacy in four models and second-most frequent in another, and "Proposal Without Verification" was also common, what are the implications of these specific failure modes for the development of more robust and trustworthy LLMs in mathematical reasoning? Are there any hypothesized architectural or training limitations that might contribute to these particular fallacies?

Similarly, the paper suggests that relying solely on final answer correctness or using powerful LLMs as judges is insufficient for evaluating logical rigour. What specific characteristics or capabilities would a more "sophisticated benchmarking method" or "improved training schema" need to possess to effectively evaluate the logical rigour and coherence of mathematical arguments?

The paper presents a breakdown of fallacy frequencies by problem type (geometry, algebra, combinatorics, number theory). Could the authors provide a deeper analysis of the unique challenges each problem type poses for LLMs and how these challenges correlate with the prevalence of specific fallacies? For example, why do geometry problems lead to more "Inventing Wrong Facts" and "Begging the Question" fallacies?

Could you provide an example of a "shortcut" used by a model that has supposedly not seen the question before, and where the problem cannot be solved by relating it to other similar questions? Specifically, what is considered to be a shortcut in such olympiad problem solvings?


Given that the current training paradigms often lack direct supervision over the sequence of actions leading to the final answer (with rewards primarily based on final answer correctness), it's plausible that the observed fallacies might be a consequence of this training setup rather than a complete lack of understanding. To what extent do you believe these findings indicate that models genuinely lack the ability to solve the questions, as opposed to merely struggling with generating human-interpretable, rigorous proofs?


The paper states that "Inventing Wrong Facts is the most frequent in four models and the second-most frequent in another... This observation might be explained by the training methods used for these models, which generally involve reinforcement learning algorithms with rewards based on the correctness of the final answers". Could you elaborate on the specific mechanisms or experimental evidence that support this connection between final answer-based reward training and the prevalence of "Inventing Wrong Facts"?

Could you clarify whether all the internal "thought tokens" generated by the LLMs were examined by the human evaluators, or only the summarized output after the models' internal thought blocks?

How were the "wrong facts" invented by the models checked and confirmed by the human evaluators to ensure that the models were indeed fabricating information rather than making subtle errors or misinterpreting established concepts?

The related work section could benefit from discussing some recent work on math reasoning robustness, such as "GSM1K" (Zhang et al., 2024), "Functional MATH" (Srivastava et al., 2024), and "Compositional GSM" (Hosseini et al., 2024). Could the authors comment on how these studies relate to or differ from the evaluation framework presented in this paper?

Modern LLMs  undergo various training phases such as pre-training (PT), supervised fine-tuning (SFT), and reinforcement learning (RL). How do the authors believe these different training stages influence the models' mathematical reasoning capabilities and the types of errors observed?

The paper highlights that the IMO shortlist problems are "highly original" and designed so that an LLM "cannot simply combine standard building blocks from well-known problems to arrive at the correct answer". However, a desirable feature for LLMs is their ability to combine standard methods and basic principles from easier problems to solve harder, more complex ones. Why did the authors choose not to include such examples in the shortlist to evaluate this specific capability?

The fallacy "Proposal Without Verification" describes instances where a method or strategy is introduced without proper justification. Could the authors provide a more detailed breakdown of how often such proposals, if pursued, ultimately lead to correct versus incorrect final solutions in the dataset?

Given that models like Gemini 2.0 and o1 exhibit a high performance in arriving at the correct final answer despite their difficulties in producing a fully correct chain of thought, what is the intuition about how these models are able to achieve accurate results without a logically sound reasoning process?

Considering these findings, what are the actionable next steps for the LLM research? How do these insights translate into concrete research questions or developmental priorities for improving mathematical reasoning capabilities?

**Reasons To Accept:**

The paper presents a robust empirical evaluation of frontier models on 455 IMO shortlist problems, prioritizing proof correctness over final answer accuracy. Human evaluators rigorously assessed solutions, lending strong credibility.

The identification and categorization of common logical fallacies in LLM generated mathematical generated solutions provides a good framework for error analysis.

The research addresses a critical gap by demonstrating that existing benchmarks, which rely solely on final answer correctness, overlook solution validity and systematic exploitation of answer predictability by LLMs. Prioritizing logical rigour, the paper advocates for more sophisticated evaluation methods essential for advancing LLMs.

**Reasons To Reject:**

The study relies on highly qualified, and therefore expensive, human evaluators. This raises some concerns. This dependency on specialized human assessment poses a significant challenge for future automation and scalability, as replicating such evaluations for newer model iterations or broader application would be difficult and costly.

The paper introduces categories for common errors observed in the solutions generated by LLMs. Although these categories offer valuable insights, the discussion lacks an exploration of their comprehensiveness, potential overlaps, or their applicability to reasoning tasks in fields beyond olympiad mathematics

The definition of a "partially correct" solution is described as one that "included some essential steps of a correct solution but omitted other crucial steps or contained significant inaccuracies", and is quite broad. This lack of specificity could make it challenging for readers to precisely grasp what qualifies as "partially correct" and to understand the consistency with which this classification was applied during the annotation process.

The authors acknowledge a high probability of data leakage, given that the problems from the IMO and IMO Shortlist are well-known and likely included in the training datasets of leading models. While it is claimed that this leakage does not significantly impact the LLMs' problem-solving capabilities, the observed low success rates could be interpreted as evidence that models are relying on recall rather than genuine reasoning. A more in-depth discussion on how this data leakage might still affect heuristic approaches, potentially leading to correct answers without sound reasoning, would be advantageous.

The experiments detailed in Section 5, which focus on the verification abilities of LLMs, seem relatively brief and could be more thoroughly developed. Considering that prior research has shown that similar advanced models can achieve better-than-chance verification performance on other benchmarks, this section appears less developed in comparison to the comprehensive analysis provided elsewhere in the paper and does not fully align with the depth of the other findings.

The paper effectively analyzes current LLM limitations in high-level mathematical reasoning but fails to adequately bridge these findings to tangible next steps for research. It lacks substantive discussion on how the insights could inform concrete questions or guidance.

Further concerns are mentioned as Questions to the Authors.

---

> ### Author Response · Authors · 2025-05-30
>
> We thank the reviewer for their careful reading and constructive feedback. We appreciate the acknowledgment of our robust empirical evaluation, the utility of our fallacy categorization, and our work's contribution to identifying a critical gap in current LLM evaluation methodologies. We address each of the reviewer's points below:
>
> ### Addressing "Reasons to Reject"
>
> 1. **Dependency on specialized human assessment:** We agree human assessment is costly and limits scalability. This dependency is a **key finding** of our work. Our goal was to rigorously assess LLM proof correctness in Olympiad math. Section 5 shows current automated methods, including LLM-as-a-judge, **cannot reliably perform this**. Human expertise remains indispensable for verifying complex mathematical proofs. Our findings highlight the **critical need for more sophisticated automated evaluation methods, e.g., theorem provers,** to address this need. Our findings also underscore the need for better approaches to training LLM for mathematical reasoning.
>
> 2. **Comprehensiveness, overlaps, and applicability of fallacy categories:** Fallacy categories were **empirically derived** from LLM solutions to IMO problems, representing **pervasive reasoning failures** in this domain. Each category highlights a distinct logical flaw, though some overlap. We acknowledge they may not be exhaustive or universally relevant beyond Olympiad math. For example, heuristics might be allowed for final answer tasks. Examples:
>
>    * **"Proposal Without Verification":** Claiming "every even natural number is a sum of two prime numbers" without proof. Its core flaw is lack of justification.
>    * **"Inventing Wrong Facts":** Citing a non-existent "Theorem 2 from 'Graphs and Random Structures' by Paul Erdos." To avoid ambiguity, **multiple fallacy labels** were allowed for consistent classification.
>
>
>
> 3. **Broad definition of "partially correct" solutions:** We agree our 'partially correct' definition needed clarity. A solution was **partially correct** if it **realized at least a high-level step with details (minor errors possible), but failed subsequent crucial steps.** Olympiad solutions can be decomposed into key steps, each with a high-level idea and derivations. Partially correct solutions realize one or more high-level steps but fail to complete all.
>
> 4. **Data leakage and its impact on heuristic approaches:** We acknowledge probable data leakage from public IMO problems. Our study focuses on **proof rigor**, not just final answers. Models' **low performance on correct proofs** indicates data leakage isn't significant for genuine problem-solving. If models recalled solutions, proof generation would be higher. Most proofs show fundamental mistakes, relying on pattern recognition or heuristics, not sound reasoning: One might find the form of a sum like $1+2+3+...+n$ by just testing examples and guessing the final pattern, but unless the guessed pattern is proven correct, it does not meet the bar as an Olympiad-level proof. Memorization affects final answers, not our proof of correctness metrics.
>
> 5. **Section 5 (Automatic Evaluation) brevity and alignment with other findings:** Section 5's brevity reflects its objective: to **demonstrate LLMs' unsuitability for verifying rigorous Olympiad proofs.** For Olympiad proof verification, a model must: 1\) **Distinguish correct from blatantly incorrect proofs**, especially with obvious mistakes. 2\) **Detect correct solutions as correct** significantly more often than labeling incorrect ones as correct. Our LLMs failed both. A simpler schema isolated this point, avoiding arguments that low performance was due to schema complexity.
>
> 6. **Lack of actionable next steps for research:** Our findings point to two critical research directions. First, **novel benchmarks** are needed to rigorously evaluate **logical soundness and step-by-step reasoning correctness**, not just final answers. High final answer performance doesn't imply solution correctness. Second, current training (e.g., RL) often rewards only final answer correctness, leading to shortcuts and hallucinations. Future research must explore **improved training paradigms** that explicitly **reward sound reasoning steps**, moving beyond insufficient LLM-as-a-judge or final-answer-only verification.

---

> > ### Author Response · Authors · 2025-05-30
> > **Responding to "Questions to Authors"**
> >
> > 1. **Inter-annotator agreement:** Consistency was ensured via a multi-stage process: shared guidelines, a meeting for shared understanding, and iterative discussion with the main author on borderline cases.
> >
> > 2. **Memorization and data leakage:** As discussed in 'Reasons to Reject,' point 4, 'pattern recognition and heuristics' refer to models getting correct final answers but failing to generate correct proofs. This includes methods like testing examples or guessing patterns without mathematically acceptable proof. Such instances yield wrong proofs, not affecting our proof correctness metrics. Memorization impacts final answer accuracy, which isn't our focus.
> >
> > 3. **Why omit essential steps?** This likely stems from LLMs' **statistical nature** trained on math data. They generate phrases like 'It's straightforward to show' due to distribution in real proofs, but lack a proper proof concept, misapplying such statements, causing this fallacy.
> >
> > 4. **Implications of common fallacies:** As discussed in 'Reasons to Reject,' point 4, these fallacies likely result from **training solely on final answers**. Models lack fundamental proof understanding, mimicking proof jargon without mathematical semantics. This highlights the need for training that rewards sound reasoning, not just outcomes.
> >
> > 5. **Sophisticated benchmarking needs:** Improving LLM verification capabilities is key, though outside our scope. A 'sophisticated benchmark' could involve a dataset of diverse proofs with varying correctness and annotated mistakes, allowing evaluation of LLM error detection within proofs.
> >
> > 6. **Fallacy frequencies by problem type:** As discussed, **geometry problems are proof-based**, leading LLMs, lacking proof grasp, to more 'Inventing Wrong Facts' and 'Begging the Question.' LLMs may struggle with spatial reasoning, inventing facts or assuming relationships without proof. Pure logical chains without algebraic expressions are harder to 'ground.'
> >
> > 7. **Example of a shortcut:** 'Shortcut' means extracting a final answer via non-proof methods that yield correct results, like 'Solution by Trial and Error.' Appendix provides examples for each fallacy.
> >
> > 8. **Lack of ability vs. struggle:** Not mutually exclusive. Observed fallacies may stem from the **training setup**, contributing to understanding deficits. Our analysis of generated proofs suggests training setup as a possible cause for this lack of understanding.
> >
> > 9. **Connection: final answer training:** The connection is based on three observations: 1\) LLMs' notable final answer performance. 2\) Their struggle with correct proofs. 3\) Different fallacy distributions between final answer and proof-based problems. This suggests final-answer-only training doesn't aid proof capabilities, leading LLMs to generate proof-like jargon without substance.
> >
> > 10. **Examination of thought tokens:** Thought tokens are generally unavailable or lack proper structure. We avoided evaluating them due to cost and time.
> >
> > 11. **Verification of wrong facts:** Expert mathematicians confirmed invented 'wrong facts' were clear fabrications violating mathematical principles, not subtle errors.
> >
> > 12. **Recent math reasoning work:** We will incorporate GSM1K, Functional MATH, and Compositional GSM into related work. Our study uses **IMO shortlist problems** for their originality, non-reducibility, multi-step nature, and conceptual depth, setting a higher bar for **proof rigor** than typical benchmarks.
> >
> > 13. **Influence of training stages:** We haven't designed experiments for this. We only discussed the possible link between post-training with final answers and observed errors. Further claims require more investigation.
> >
> > 14. **Why not standard methods?** We chose shortlist problems to prevent models from reducing them to famous problems using vast knowledge. This measures real problem-solving, not just knowledge vastness, highlighting shortlist originality.
> >
> > 15. **Proposal verification frequency:** We didn't gather this data due to high cost and time. It requires evaluators to assess if a proposal works or is plausible, but failing to reject plausibility doesn't guarantee a viable solution.
> >
> > 16. **Intuition: correct answers without logic:** As discussed, models achieve this by exploiting shortcuts like testing small examples or trial and error, without generating sound proofs.
> >
> > 17. **Actionable LLM research steps:** As discussed in 'Reasons to Reject,' point 6, our findings highlight two key actionable steps: focusing on proof-oriented benchmarks and developing improved training paradigms rewarding logical rigor beyond final answers.

---

> ### Comment · Reviewer_qtx8 · 2025-06-09
>
> Thank you for the detailed response.  I appreciate you taking the time to address the concerns raised. While your clarifications are helpful, I find that several of my original points of concern remain insufficiently addressed.
>
> My concern about human evaluation still remains. The limitations of LLM-as-a-judge or similar automated methods are generally known (lots of prior work on this). Stating that the dependency on expensive human evaluation is a "key finding" of your work seems illogical. Also, framing a methodological limitation as a primary finding feels like a post-hoc justification. So the core issue remains that the scalability and replicability of your own analysis are severly constrained.
>
> With flaw categories overlapping it is hard to pinpoint and control why the model made a certain mistake.
>
> I find the argument about data leakage unconvincing. The concern is not necessarily that models are recalling entire proofs (or generations), but it is about exposure to the problem set (or similar problems with, for instance, on policy learning methods).
> The connection between leakage, heuristics, and the observed fallacies is not clear or very convincing.
>
> You suggest that your work points toward the need for "novel benchmarks" and "improved training paradigms." While correct, these are generic recommendations applicable to nearly any paper in the field.
>
> Regarding the thought tokens (when available), This seems like a critical omission. The distinction between a model lacking the ability to reason and a model struggling to articulate a rigorous proof is fundamental. Analyzing the chain-of-thought, however unstructured, is essential for probing this distinction.
>
> Regarding the intuition for correct answers without logic, your reply is plausible, but feels like an oversimplification for many abstract problems in the IMO shortlist that do not have simple, guessable numerical answers.
>
> Thank you again for your response.

---

> > ### Author Response · Authors · 2025-06-10
> >
> > Thank you for your thoughtful follow-up and continued engagement with our work. We appreciate your persistence in raising these important concerns, and we'd like to address each point in detail.
> >
> > **Concern about Human Evaluation:**
> >
> > While the limitations of LLM-as-a-judge methods are generally known, our work reveals that these methods perform essentially randomly on high-school contest problems such as Olympiad mathematics, a finding that was not previously established. Given that we demonstrate LLM-as-a-judge approaches yield random results, human evaluation becomes necessary at this stage. As an intuitive requirement, a model should be able to distinguish correct proofs from blatantly incorrect ones. Our results show that current models lack this fundamental capability
> > While scalable evaluation methods would certainly be preferred over human evaluation, until the problem of LLM randomness on proof assessment is solved, human evaluation remains the only viable option.
> >
> > **Overlap Between Fallacy Categories:**
> >
> > Although we acknowledge potential overlapping cases, each fallacy occurs under specific circumstances and in particular situations, as our analysis and plots demonstrate. The patterns we observe suggest distinct categories despite some boundary cases.
> >
> > **Concern About Data Leakage:**
> >
> > We do not claim a direct causal link between data leakage and the observed patterns. Our high-level message is that there is no correlation between proof correctness and final answer correctness, and the latter is not a reliable proxy for the former. Our human evaluations reveal systematic patterns in these generated solutions. We suggest there might be connections between post-training methods and these heuristics, but this remains speculative.
> >
> > Importantly, there are differences in mistake patterns between problems with concrete final answers versus those without. If leakage were the primary cause of observed heuristics, we would expect to see partially correct proofs frequently across different problems, but all models struggle consistently with proof generation. The percentage of partially correct proofs is pretty low for all models, so these models do not remember even a single piece of non-trivial, useful information, in case of any possible leakage.
> >
> > The existence of data leakage does not undermine our work. With or without leakage, these models are trained to find final answers (based on publicly available information about post-training), which fundamentally disregards proof correctness and allows for heuristic-based approaches to problems with concrete answers.

---

> > > ### Author Response · Authors · 2025-06-10
> > >
> > > **Concern About the Paper's Contribution:**
> > >
> > > Evaluating models' reasoning capabilities using mathematical benchmarks and checking final answers remains one of the most common methods in LLM reasoning evaluation. We believe our work provides strong motivation for developing more sophisticated evaluation frameworks in the future.
> > >
> > > **Concern About Chain-of-Thought Token Evaluation:**
> > >
> > > We believe this concern is not well-founded. If the issue were merely that models can solve problems but struggle to articulate solutions, they should perform well on binary classification of correct versus incorrect proofs. Since this is not an articulation problem, models need only choose between outputs, but we observe performance close to random; this suggests a deeper limitation in mathematical reasoning capability.
> > >
> > > **Concern About Intuition for Correct Answers Without Logic:**
> > >
> > > We observe a substantial discrepancy between final answer correctness and proof correctness, with two key observations:
> > >
> > > 1. This gap is large across all evaluated models.
> > > 2. Despite this large gap, the best model achieves only 63.2% final answer accuracy, calculated specifically among problems with concrete final answers.
> > >
> > > While many abstract problems in the IMO shortlist lack simple, guessable numerical answers, this claim applies specifically to problems with concrete final answers. Finding final answers is generally easier than constructing complete proofs. Many IMO shortlist problems do have final answers that are not easily guessable, but guessable answers are common since IMO is not multiple-choice, and typically, finding only the final answer receives zero points in proof-based contest rubrics.
> > >
> > > There are many cases where finding final answers is relatively straightforward:
> > >
> > > 1. Functional equation problems where solutions can be found by testing simple functions.
> > > 2. A common class of problems requiring proof that no non-trivial solutions exist, where the main challenge lies in proving non-existence rather than finding the answer itself. Diophantine equations often fall into this category.
> > > 3. Combinatorial problems asking whether a described process can achieve a desired outcome, which have yes/no answers but require substantial proof of possibility or impossibility.
> > >
> > > This explains why "proof by example" and "solution by trial and error" appear significantly more frequently in problems with concrete final answers compared to purely proof-based problems.
> > >
> > > We hope this clarifies our position and addresses your concerns. Thank you again for your valuable feedback, which helps strengthen the discourse around evaluation methods in mathematical reasoning.

---

### Official Review · Reviewer_1MfA · 2025-05-14

**Rating:** 8
**Confidence:** 4
**Ethics Flag:** 1

**Summary:**

This paper presents the result of evaluating LLM-generated answers to Math Olympiad short-list problems.
Typical fallacies in the answers are classified to six categories and experts in math or CS annotated the LLM-generated answers with those categories.
As a result, it was revealed that state-of-the-art LLMs generate correct answers to less than 4% of the problems.
The types of fallacies are fairly similarly distributed across the evaluated LMMs.
In addition, by testing whether the LLMs can discriminate between fully correct answers and those with fallacies, it was revealed that those LLMs' judgements are almost random, which suggests that it is inadequate to employ LLM-as-judge for the Olympiad-level math solving.

**Questions To Authors:**

Do you have any plan to make the annotated dataset public?

**Reasons To Accept:**

- The paper exposes a significant fact about the ability of current LLMs on high-level math problem solving
- The writing is clear

**Reasons To Reject:**

- I found no significant reason to reject this paper

---

> ### Author Response · Authors · 2025-05-30
>
> We thank the reviewer for their positive assessment, recognizing the significance of our findings regarding LLM capabilities in high-level mathematics and the clarity of our writing. We appreciate the confidence in our work.
> Regarding your question about making the annotated dataset public: Yes, we do plan to release the annotated dataset publicly. We are currently working on adding more comprehensive annotations to the data and aim to make it available by the end of summer.

---

> > ### Comment · Reviewer_1MfA · 2025-06-10
> >
> > Great to hear that the dataset will be made public with additional annotations!

---

### Decision · Program_Chairs · 2025-07-08

**Decision:**

Accept

**Comment:**

This paper critically examines reasoning LLMs ability to solve Olympiad-level mathematical problems and show that they often fail to produce rigorous and logically sound proofs. While LLMs can occasionally arrive at correct final answers, these are frequently the result of pattern recognition or heuristic shortcuts rather than genuine mathematical reasoning. Their qualitative and quantitative evaluations underscore a substantial gap between LLM performance and human expertise in advanced mathematical reasoning, emphasizing the need for benchmarks that prioritize the rigor and coherence of mathematical arguments over just answer correctness.

Overall, I favor acceptance and request the authors to address the concerns raised by the reviewer (particularly qtx8)

Potential areas for improvement:
(1) The automatic eval section lacks mention of some important details, should contain verification accuracy (false positive and true positive rates) as a function of inference time compute.
(2) Evaluating LLMs CoT reasoning on existing Olympiad benchmarks with concrete final answers for comparison. Are the limitations of LLMs reasoning correctness mostly on proof-based problems or also on problems with a concrete final answer.